# Multi-Variant Damage Assessment in Composite Materials Using Acoustic Emission

**DOI:** 10.3390/s25123795

**Published:** 2025-06-18

**Authors:** Matthew Gee, Sanaz Roshanmanesh, Farzad Hayati, Mayorkinos Papaelias

**Affiliations:** School of Metallurgy and Materials, University of Birmingham, Birmingham B15 2TT, UK; mwg956@student.bham.ac.uk (M.G.); s.roshanmanesh@bham.ac.uk (S.R.); f.hayati@bham.ac.uk (F.H.)

**Keywords:** acoustic emission, fibre-reinforced polymers, structural health monitoring, digital signal processing, multi-variant, Fourier transform, wavelet transform, k-means, DBSCAN

## Abstract

**Highlights:**

**What are the main findings?**
The accurate identification of damage modes in composite materials through the analysis of the frequency content of acoustic waveforms.A multi-variant analysis of acoustic emission signals using fast Fourier transform and wavelet transform analysis based on the empirical wave transform.

**What are the implications of the main findings?**
Quantitative acoustic emission based on the frequency range of signals.The classification of acoustic emission signals with respect to damage modes.

**Abstract:**

This study presents a novel methodology for the real-time characterisation and quantitative assessment of damage in fibre-reinforced polymers (FRPs) using acoustic emission (AE) techniques. While FRPs offer superior mechanical properties for structural applications, their anisotropic nature introduces complex damage mechanisms that are challenging to detect with conventional inspection methods. Our approach advances beyond traditional peak frequency analysis by implementing a multi-variant frequency assessment that can detect and evaluate simultaneously occurring damage modes. By applying the fast Fourier transform and examining multiple frequency peaks within AE signals, we successfully identified five distinct damage mechanisms in carbon fibre composites: matrix cracking (100–200 kHz), delamination (205–265 kHz), debonding (270–320 kHz), fibre fracture (330–385 kHz), and fibre pullout (395–490 kHz). A comparative analysis with wavelet transform methods demonstrated that our approach provides earlier detection of critical damage events, with delamination identified approximately 28 s sooner than with conventional techniques. The proposed methodology enables a more accurate quantitative assessment of structural health, facilitating timely maintenance interventions for large-scale FRP structures, such as wind turbine blades, thereby enhancing reliability while reducing operational downtime and maintenance costs.

## 1. Introduction

### 1.1. Background and Requirements

The increasing use of fibre-reinforced polymers (FRPs) for the manufacturing of critical components in demanding environments has created a need for reliable online assessment methods based on structural health-monitoring (SHM) techniques. Compared with metals and alloys used in advanced structural applications, lightweight FRPs often demonstrate superior performance coupled with excellent mechanical properties, whilst they are immune to corrosion and stress corrosion cracking (SCC). The mechanical properties of FRPs arise from their microstructural characteristics, which incorporate either unidirectional or multidirectional fibre configurations embedded within a polymer matrix. Hence, their properties and performance can be tailored to meet specific application requirements [1]. However, these unique microstructural characteristics also introduce anisotropy and complex damage mechanisms, rendering inspection challenging.

FRP components are inspected using non-destructive testing (NDT) techniques capable of detecting and evaluating different types of defects and their severity. One commonly used NDT method for the inspection of FRPs is ultrasonic testing (UT). UT enables the accurate detection and localisation of defects and damage within FRP components [2]. A key limitation of this technique is that the sensitivity and resolution of the inspection are dependent on the frequency and size of the probe used, and thus, defects smaller than half the wavelength of the interrogating ultrasonic beam cannot be detected at all. Moreover, damage can only be detected during inspection and not while the component is in operation, hence, limiting the opportunities during which UT can be carried out. Since conventional UT can only be applied offline, a lower availability of the asset concerned may have to be accepted.

With conventional NDT methods, damage can only be evaluated periodically. Thus, predicting how any damage detected may progress between inspection intervals will be necessary. Predicting damage growth in FRPs is not straightforward, and approaches such as the Paris–Erdogan law and the Palmgren–Miner linear damage rule do not usually work well. Hence, it is not always possible to achieve sufficient accuracy in predicting structural degradation in these materials. Wind turbine blades (WTBs) are large-scale industrial FRP structures installed in remote locations, which are often difficult to access. Manual inspection, apart from requiring a shutdown of the wind turbine, can be affected by limited accessibility due to weather conditions. WTBs are subject to stochastic loads and adverse environmental conditions. This means degradation can accelerate unexpectedly due to occasional overload events caused by powerful wind gusts. Once severe damage is detected, repair or replacement of the affected WTB will need to be scheduled based on the availability of necessary spare WTBs and lifting equipment. Hence, real-time SHM offers distinct advantages over conventional inspection methods. Apart from allowing the continuous evaluation of the structural health of WTBs, it can contribute to the effective planning and scheduling of the maintenance intervention required.

The implementation of a reliable SHM system can help ensure the reliability of large FRP structures, including WTBs, enabling accurate, effective, and cost-efficient maintenance planning. Acoustic emission (AE) is a passive technique that can be used for the effective SHM of composite structures. However, the characterisation and quantitative evaluation of the severity of damage detected using existing approaches for AE data interpretation have significant limitations. Within the present study, we report on the development of a novel methodology for the accurate real-time characterisation and quantitative assessment of damage detected in composite structures using AE.

### 1.2. Current Approaches

The monitoring and evaluation of FRP structures using AE relies on the use of digital signal-processing (DSP) techniques to extract features from the recorded signals. These signal features can subsequently be used to evaluate the characteristics of the damage detected. There are several methods which can be applied to achieve this, although the accuracy can vary significantly depending on the one chosen. In the methodology studied here, frequency components in the AE signal recorded can help increase accuracy in the evaluation of the severity and characteristics of the damage events detected. Unlike other parameters which can be extracted from an AE signal, such as the peak amplitude, AE energy, root mean square (RMS), rise time, etc., the effects of the acquisition settings, together with the amplification used and the distance between the sensor and source, can influence the values recorded, affecting the conclusions drawn, however, this is not the case for frequency.

Direct damage assessment can be achieved through the evaluation of the input signal in the frequency domain, together with the relevant frequency components. A common way to do this is through the evaluation of the peak frequency value of a captured AE event. Frequency values can be used in different ways to evaluate the damage mode that has occurred. There are various mechanical tests that can be used to cause damage in FRPs to evaluate the associated peak frequency value. One approach involves tests that cause individual damage modes to occur through specific experimental setups, such as matrix cracking caused during the mechanical testing of a pure resin system [3,4]. Such tests ensure the observation of the intended damage mode. However, they do not account for any changes resulting from the structural differences and the effects of the constituent components, i.e., resin and fibres. Alternatively, mechanical testing can be carried out on FRP specimens where no target damage mode is specified, and, instead, the aim is to replicate a realistic loading pattern for a component or material [5], such as tensile testing for a tensile–tensile loading system or flexural testing for a tensile–compressive system. The peak frequency, amplitude, and energy of the AE signals obtained during such tests must typically be correlated with the damage sustained by simultaneously evaluating the stress–strain curve.

### 1.3. Main Limitations of Current Approaches

The type of sensor employed affects the sensitivity of the acquisition for particular frequency ranges, particularly in the case of resonant sensors. However, with the more recent developments of flat-frequency-response AE sensors, this is less of a concern. However, the use of flat-response-frequency AE sensors can influence the effect of background noise in certain cases, depending on the operational range of a particular sensor. The sampling rate is an additional important parameter, which should be ensured to be selected on the basis of the operational frequency range of the sensor and the frequency range of the expected signal acquired during the measurement. Where possible, appropriate filtering methodologies should be employed to minimise or eliminate the effects of external noise sources that can make effective damage characterisation and quantification more challenging. The material dependencies of the analysis should be accounted for in different FRP systems, i.e., the type of resin and fibre employed.

Additional limitations include the type of analysis carried out, whether based on statistical or waveform analysis methods. Subsequently, the parameters used, together with the acquisition settings, can play a significant role in the assessment of damage severity using AE techniques. The effects of the level and type of background noise can be less or more significant in the analysis carried out.

### 1.4. Frequency-Based Analysis

FRPs exhibit anisotropic mechanical properties, which are strongly dependent on the fibre orientation. The fibre orientation and anisotropy characteristics of FRP materials introduce an additional level of complexity when studying the transmission of elastic stress waves through the material [6]. Taking into consideration how an individual elastic stress wave is generated at the crack tip as it propagates and attenuates within the structure, frequency-domain-based analysis becomes advantageous over time-domain-based evaluation.

The propagation of Lamb waves in plate structures can be characterised by the phase (*c_p_*) and group velocities (*c_g_*). The group velocity refers to the entire wave package and the velocity at which it propagates within the medium. However, individual phases of different frequencies, and hence, different velocities (*c_p_*), exist within a wave package and its constituent group. Let us consider a wave package emitted from an event within the material, such as matrix cracking combined with fibre fracture, which excites the surrounding particles at two distinct energy levels, hence producing two distinct frequencies. For this individual wave package, containing varying frequency components from the same source, the detection and evaluation of these is possible, providing the ability for their simultaneous assessment. This is due to the dispersion of the associated phase velocities (*c_p_*) within the overall group velocity (*c_g_*); hence, the arrival time of these components will propagate faster or slower, accordingly. Dispersion allows for each of the phases to be individually assessed, as there is limited overlapping or combination when considering the captured signal [6,7].

A multi-variant-based methodology for the analysis of AE signals through which the simultaneous occurrence of multiple damage modes can be assessed and quantified is of significant interest. Using the proposition of discrete frequency ranges that have been attributed to specific damage modes, and the separation of frequencies within a wave package due to dispersion in the *c_p_*, the designation of more than one frequency for the description of an AE event is required. By taking into consideration the peak frequency rather than the signal amplitude, we can distinguish the difference of damage evolution, including, initiation, matrix cracking and propagation to later stages including, delamination, and final failure. In conventional analysis of AE signals, delamination-related events can overlap with extensive matrix cracking, which would influence the interpretation accuracy of the structural health of the structure significantly.

The consideration of the frequency spectra beyond the observed peak frequency, as suggested above, has been explored to some extent in previous studies [3,8]. This is achieved through the use of the fast Fourier transform (FFT) on an extracted waveform and an in-depth assessment of the produced frequency domain representation. Fotouhi et al. demonstrated the presence of multiple peaks in the frequency spectra related to multiple damage types for a single damage event [8]. This was limited, however, by its lack of expandability, focusing only on a few damage events. Similarly, the labelling and attribution of further peaks within a produced frequency spectrum for a damage event was completed by Bussiba et al. for the rationalisation of the damage processes seen within an FRP [3]. However, the work completed within the referenced literature was not completed upon the entirety of the damage events from throughout the monitored tests, and, instead, it was applied to a few examples only.

The use of wavelet transforms (WTs) has been studied to assess the frequency components of a given AE signal. WTs were explored by Qi to determine damage in FRP materials [9]. Barile et al. focused on a discrete wavelet transform (DWT) for the evaluation of the relative energies of the spectral components within an input waveform [10], essentially evaluating the energy distribution within the frequency spectrum of the AE signals. Within this study, the frequency ranges assigned overlap with the frequencies assigned to different damage modes.

The aim of the technique studied herein is to allow for the accurate characterisation and evaluation of the severity of different forms of damage occurring within FRP structures. To implement a quantitative AE system, accurate characterisation and evaluation of the severity of damage modes are required.

### 1.5. Other Types of Materials and Additional Considerations

The principles and ideas outlined within the Introduction, as well as the completed research throughout this study, have been strictly completed within the context of application to FRPs. Their behaviour with respect to the transmissibility of AE signals (see Section 1.4) requires this focus. However, it is important to note that this work should not be considered as applicable exclusively to FRPs. There are other materials where AE is implemented in their monitoring, where the explored concepts may be applied. For areas where the evaluation of specific mechanisms of discrete energies is key, such as in stress corrosion cracking, fatigue cracking, etc., the application of the proposed methodology may be considered. Rather et al. highlighted the importance of the techniques discussed within Section 1.2 within the monitoring of concrete for both torsional loading as well as corrosion [11]. Muñoz-Ibáñez et al. presented the use of similar techniques for the monitoring of mode I fracture toughness tests on sandstone rocks, with other works completed regarding the assessment and monitoring of masonry structures [12,13]. Additionally, WTs have been applied to the monitoring of asphalt, as presented by Xu et al., where the work completed here may serve as a viable alternative for a more in-depth study or an extended monitoring case [14]. The examples given above are not exhaustive regarding the possible applications or alternative materials where this may be viable; however, they begin to provide insight into how the extension of this work is feasible given the correct criteria.

## 2. Materials and Methods

The materials and equipment used in the study are outlined in the following sections.

### 2.1. Materials

The test carbon fibre-reinforced composite (CFRC) coupons used in the present study had a layered laminar structure and were manufactured using the resin infusion process as part of the Horizon 2020 Carbo4Power project. The CFRC material exhibited a Young’s modulus of 1.20 × 10^11^ N/m^2^. The tensile coupon samples tested were 250 mm long with a width of 25 mm and a thickness between 1.03 and 1.33 mm, with a layup of [0/45/QRS/−45/90]. Quantum resistive sensors (QRSs) were embedded within each sample to monitor strain, temperature, or humidity. The samples were end-tabbed using resin. The sample coupons for flexural testing had a length of 150 mm, a width of 25 mm, and a thickness of 1.15–1.32 mm, with a layup of [0/QRS/45/−45/90], with no end tabbing required.

### 2.2. Mechanical Testing

Mechanical testing was carried out to collect the required data used for the present study, as well as to qualify the materials for use in the manufacturing of the demonstration blades planned within the Carbo4Power project. Tensile and flexural (3-point bending) tests were carried out on the available samples.

Tensile testing was carried out using a Zwick-Roell electro-mechanical testing machine (model number: 1484), with a 200 kN load cell, a crosshead movement speed of 2 mm/min, and the extensometer set at 10 mm. Flexural testing was completed using a Dartec Universal Mechanical Testing Machine with a 50 kN load cell and crosshead movement speed of 1 mm/min. Ambient laboratory testing environmental conditions were applied for all tests.

### 2.3. Acoustic Emission Acquisition

During the mechanical testing, a customised 4-channel AE acquisition system built by the authors at the University of Birmingham was used. AE activity was detected using an R50α resonant sensor, with an operational frequency range of 100–700 kHz [15]. The AE sensor was mounted on the sample surface using Araldite^®^ made by Huntsman Advanced Materials in the United Kingdom, a two-component epoxy adhesive, which also provided the ultrasonic coupling. The output from the AE sensor was subject to a pre-amplification and a main amplification stage. Pre-amplification was achieved through a PAC 2/4/6 pre-amplifier set at 40 dB. The main amplification was achieved using a Feldman Enterprises Ltd. 4-channel amplifier, with the amplification set at 9 dB. The AE signals were filtered at 500 kHz for anti-aliasing purposes and digitised using a 4-channel 16-bit National Instruments 9223 Compact Data Acquisition (NI-cDAQ) module. The NI-cDAQ module was connected to a compact Intel Next Unit of Computing (NUC) computer using a cDAQ-9171 connected via a USB port. The data acquisition was performed using custom-written software in MATLAB v. 2025a and developed in-house using NI-DAQmx running on the NUC computer. The recorded datasets were captured at 1 MSamples/s per channel periodically for 5 s with a 1-second interval between each recording. Figure 1 shows the mechanical testing set up with the AE sensors attached for both tensile and flexural testing. The schematic diagram in Figure 2 shows the overall system architecture adapted from [16].

### 2.4. Digital Signal Processing

The nature of the data captured using the customised acquisition system required a bespoke data-handling and DSP approach to translate the raw captured AE signals into usable information. This involved all necessary processes, including importing and data handling, preprocessing, and signal correction, the location of damage events and the extraction of relevant data, the application of required signal-processing techniques, the identification and extraction of important information, and, finally, the visualisation of the data completed with MATLAB-written software.

Several processes and operations are required in a DSP system. Herein, we consider only the key ones applicable. One of these key processes is the correction of the DC offset introduced to the signal during acquisition, using a defined threshold to prevent any unwanted electromechanical noise captured within the background of the signal from being identified mistakenly as a damage event. However, for different materials, the selection of a single stationary value that would be suitable for all of them is not possible due to the different acoustic impedances applicable. Hence, dynamic evaluation of an appropriate value based on the content of the signal has been introduced. This allows for adjustments to be made based on the material being evaluated to account for any deviation in noise, as well as the prioritisation of the selection of damage-related information contained in the signal. Following this, AE signals related to damage events can be identified, evaluated, extracted, labelled, and time-stamped accordingly. Once these have been identified for the input signals, the desired DSP technique, i.e., the Fourier transform using the FFT algorithm, can be applied.

After this transformation, for both the peak frequency and multi-variant methods to be explored, there is a need to employ further peak selection. For the peak frequency, only the maxima of the spectra need to be selected. However, for the extraction of multiple peaks, restrictions and selection parameters must be applied to ensure the accuracy of the extracted values as well as prevent the selection of overlapping values.

## 3. Results

### 3.1. Peak Frequency Assessment and Damage Mode Classification

For the characterisation of the materials tested and damage evolution during testing, frequency-based analysis can be used. In the present study, the peak frequency of AE signals related to damage events was evaluated. To maximise the effectiveness of this technique, k-means clustering was applied to evaluate the signal clusters arising as damage evolved and extract the values of these frequency bands. In this way, damage events associated with different damage modes can be related to different frequency ranges. Due to the nature of k-means clustering, some level of operator input and level of insight is required. This includes the number of clusters to be assigned, which are required to be specified in advance of the application of the technique [17]. The number of clusters typically assigned should be equal to the expected number of damage modes observed in the FRP structure, including matrix cracking, matrix–fibre debonding, interlaminar delamination, fibre fracture, and fibre pullout [18,19]. However, in practice, this is not always sufficient, as variations in the aforementioned damage mechanisms can result in signals assigned to a single cluster. For example, transverse and longitudinal matrix cracking are two different mechanisms resulting in different AE signal intensities [20]. As the energy of an event and frequency can be directly related to each other, it is feasible to attribute separate bands to each of these specific mechanisms in order to differentiate them [21,22]. Multiple applications of the k-means clustering algorithm on the same dataset were additionally performed to ensure that the proper assignment of the clusters was achieved and to prevent misidentification [23]. The likelihood of misidentification can be improved through the control of the input data itself, as k-means is sensitive to outliers within the dataset. In this regard, points of lower magnitudes were filtered through the implementation of a dynamic threshold based on the input data (see also Section 2.4). This contributes to clearly defined clusters being established. However, the shape of the clusters being defined is not inherently suited to the application of k-means, where spherical clusters are assumed, and, thus, may result in the poorer sensitivity of this technique. Depending on the way that clusters are assigned, for example, applying random cluster centroids can result in misidentification [17]. The effects of this can be limited through the selection and control of the random number generation within the chosen software, here MATLAB, helping improve repeatability in the initial seed or centroids selection. However, this can limit the reproducibility of the results, such that comparison with hand-labelled data is required to ensure proper implementation.

For the assignment of the number of clusters (k) there are different methods which are suitable, such as the elbow method or silhouette score. Herein, both were used in series with each other. Firstly, the elbow method for the value of k was employed, and then the silhouette score was used to ascertain the suitability and accuracy of clustering with respect to the number of clusters. As the complete dataset includes a variety of results, from different test types, it should be considered non-uniform in its information distribution, hence the suitable k. Due to this, an assessment regarding the suitability for k was completed for each of the individual tests, separately. Figure 3 provides an example of the application of the elbow method, using the sum of the squared distance measure for points within a cluster, to provide information on a suitable cluster number [24].

The point of inflexion, or elbow, was selected, shown here as 6. However, a criticism of this technique is the subjective nature of its location. Therefore, for completeness, in this example, k-values of 4, 5, 6, and 7 were tested, using both the silhouette score and visual assessment of the clusters. The colour assigned to a given cluster is representative of the order in which the clusters were assigned.

Figure 4a–d shows both the application of k-means for a set value of k, as well as the respective silhouette score for the assigned clusters. In a quantitative capacity, the silhouette score is of greater use, where the closer it is to 1, the better the assignment of the cluster that point has [25]. For numerical assessment, the mean score across all clusters can be used, with these being 0.9209, 0.9300, 0.9314, and 0.8802, respectively, for k-values of 4, 5, 6, and 7. From this, it can generally be assumed that the overall quality of fit of any point to an assigned cluster is greatest when k is 6 for this example dataset. This can be corroborated visually, in a more qualitative manner, through the visual inspection of the shape of the silhouette for each cluster, ensuring that the values for each point within it are above a given threshold and are uniform throughout, unlike cluster 1, when k is 7, where this is not observed. Additionally, the clusters themselves, as well as the information which can be extracted from them, can also be assessed, in this case, the values for the extracted frequency bands. Herein, the greatest disparity was seen in those assigned within the centre of the frequency spectra, experiencing either the splitting or combination of bands, greatly affecting the evaluated ranges. For consistency, the successful clustering is displayed in Figure 5, following the same colouring scheme used throughout the remainder of the manuscript.

Figure 5 demonstrates the relationship between the damage modes observed within the material and how the assigned clusters from Figure 4c correspond to these. The links between the evaluated clusters, their respective ranges, and the damage modes require operator insight. However, in the extraction of the applicable frequency ranges of a cluster, the numerical sorting helped align each cluster with the corresponding damage mode.

The accuracy with which these ranges are assigned can be further improved through the introduction of an additional clustering technique. This should be, preferably, an unsupervised technique where cluster number designation is not required. A technique that meets this criterion, as well as being better suited to the data, their dimensionality, and their inherent cluster shapes, is density-based clustering, specifically the DBSCAN algorithm. The outlined criteria, which DBSCAN meets, by Ester et al., specifically the suitability for large databases and clusters of arbitrary shapes, are highly desirable considering the test data [26]. To summarise, DBSCAN defines clusters based on a defined neighbourhood (ε) around a point and a defined minimum number of points (minpts). A given point can be designated as a core point, one where the surrounding ε is populated by at least the minpts, being density-reachable. If a point does not meet this criterion, it can be designated as a border point, provided it is directly density reachable from a core point, or, if this is not the case, it is designated as noise, excluded from any clusters formed through this method [26]. Acceptance of noise presence is also favourable for the given dataset, forcing points that are outliers to fit into a predetermined cluster.

In terms of its application here initially, a minpts value must be decided, and this has been argued to be more arbitrary, provided it exceeds the dimensionality of the input data [27]; therefore, this was scaled accordingly relative to the input dataset. For the tensile example chosen, a value of 15 was used. From this, a value for ε could be evaluated using a k-distance elbow plot, evaluating the distance between neighbours using a specified distance measure. For the distance measure, testing was completed for the suitability of each for the specified dataset. Of these, it was found that the Manhattan distance measure best suited the elongated clusters observed. With this, ε could be evaluated, and the technique was applied for comparison against the k-means and hand-labelled ranges.

Figure 6a,b demonstrates the complete methodology for the application of DBSCAN to the dataset. Whilst it still required some operator input, for the evaluation of ε, the designation of the number of clusters being decided relative to the input data and parameters offset the possible negative aspects of k-means, making this a complementary supplementary technique, especially considering the inconsistencies in the number of observable bands between samples. From the evaluation of the extracted ranges from the established k-means and DBSCAN approaches, compared with the hand-labelled data, the following ranges were assigned to the damage modes.

In Table 1, the two identified mechanisms for matrix cracking have been combined into a single band for simplicity, as well as having a lower threshold limit applied on this band despite data points existing below 100 kHz, as can be seen in Figure 1. Points detected within the frequency range of 50–100 kHz fall outside any established trend, as well as being outside of the optimum sensitivity of the sensor being used, i.e., 100–700 kHz [15]. The values in Table 1 are an overall approximation based on the complete dataset involving both tensile and flexural test results, with the listed colour of how they will be shown in representations with multiple damage modes. It should be noted that not all samples displayed every possible damage mode listed when using the peak frequency assessment method. In such cases, a null value was entered and not included in the final evaluation. The specific values collected herewith for the range frequency band of each damage mode are difficult to directly compare with the available literature [28], as these values are a product of a specific material and its properties, alongside the customised AE setup employed, particularly, the sensors used. However, the general ordering and their position relative to the frequency spectrum do align with the literature values [3,19,29], as demonstrated in Table 2.

Finally, there was a need to physically confirm the occurrence of these identified damage modes within the completed mechanical testing to validate the above findings. The specific damage modes were confirmed using scanning electron microscopy (SEM) to visually confirm the occurrence of these damage events. However, in the work completed here, this approach was limited, as this was only completed for the post-test assessment. Accounting for this, it can only be used in order to confirm that they did occur at some point during the testing, but not when. Hence, it cannot be used to directly validate the presented AE information, as they cannot be matched to specific occurrences. This would be an area of improvement if this step was introduced into the methodology to directly confirm the occurrence of damage and the identified response in the specific sample. An example of a tensile-loaded specimen can be observed below, showing the fracture site.

However, despite the shortcomings of SEM used in this way, as presented in Figure 7, there are alternative methods for the confirmation and validation of specific damage modes. One damage mode that can be confirmed through traditional methods is delamination. This can be achieved through the comparison of the AE data, as a time series, against the loading curve from mechanical testing. For the considered tensile example, this can be evaluated through the departure from linear loading [4,8]. In this manner, the frequency region extracted can be confirmed through the location of this event and its relation to the observed frequency response. Similarly, flexural testing can yield similar results through the designation of areas within the loading trace.

Figure 8 shows the loading trace represented against time, with the isolated peak frequency values for the delamination region shown also relative to time. It should be noted that for the time series, there was a discontinuity between the loading and AE timings, as these systems were not directly linked, resulting in a lack of synchronisation in their starting. Shown in Figure 8 is the first major delamination, as mentioned earlier; however, more instances can be observed upon closer inspection of the loading trace, yet the example provided, demonstrating the technique and the link between the first major delamination and the first delamination peak, is sufficient.

### 3.2. Multi-Variant Assessment

The main drawback associated with the use of the peak frequency for damage assessment in practice is the lack of consideration for damage, which is happening simultaneously, but at a lower intensity level. Even when a particular damage mode is not dominant, at least in terms of the spectral representation, it is still vital that it is evaluated as it may provide insight into the initiation of damage, allowing for its early detection and maintenance intervention to be carried out promptly. A simple yet effective means of implementing such a system involves analysis of the entire frequency spectra for each identified event within a captured AE signal. There are several ways in which this can be achieved, as explored within Section 1.3 of this paper. Within the present study, two methods were considered. The main focus was on the use of the Fourier transform, but with the evaluation of the entire frequency spectra. The WT, as discussed by Qi (2000) and Barile et al. (2022) [9,10], was evaluated as an alternative, even though this was not the main focus of the present study.

#### 3.2.1. Fourier Transform

The assessment of the peak frequency of a given signal involves the isolation and extraction of damage events. The FFT can be applied to transform the data into its frequency domain and extract the peaks of interest, under a set of strict selection parameters, to ensure the choice of independent peaks of significant value, as detailed in Section 2.4. This also involves the selection of the maximum number of peaks that are to be extracted for each of the input signals. In the present study, five peaks were selected in relation to each of the damage modes identifiable. To prevent desynchronisation, as well as for the ease of representation, any inputs which did not yield significant peaks to trigger their acquisition were filtered out using null values for the uniform length of each level.

Figure 9a,b shows the established selection parameters based on the frequency domain. It also shows the distribution of the signal within the frequency spectrum, allowing for comparison with the result from the WT performed on the same input signal. As observed, only three peaks met the selection criteria established. Thus, these were recorded in the highest three of the levels, and null values were inputted into the lower two.

For the evaluation of the information, proper representation was required. This allowed for distinguishing between the presented levels within the data, as well as the establishment of any apparent trends. To best display this, the division of each magnitude level relative to other peaks within the same event was performed, each into a single plot. Each of these was further subdivided based upon the frequency ranges established earlier in the investigation using the peak frequency values and k-means clustering, represented by the assigned colour scheme.

Figure 10a,b demonstrates the discussed technique applied to individual examples from both the (a) tensile and (b) flexural datasets. The plots in Figure 10a,b show, from left to right, the different intensity levels extracted from the frequency spectra of the damage events identified within the inputted examples. Here, the intensity levels refer to the relative magnitude of each of the extracted peaks from a single identified damage event, such that level 1 refers to the peak of the greatest magnitude, and level 5 to that of the least magnitude. Within each of the produced subplots within Figure 10a,b, the colour used for the points represents the associated damage mode, given in the legend, related to the associated frequency ranges in Section 3.1. Despite the large input data sizes, the operations performed to acquire such data were efficient timewise for the computer model used for processing, a Dell Inspiron 16 7610 (Intel 11th Gen i7-11800H with an upgraded 32 GB RAM at 3200 MT/s, with up to 14 GB allocated to MATLAB). For the tensile test data, around 220 million input data points (220 s) were evaluated for each sample, yielding around 6000 damage events identified and, hence, resulting in a total of 30,000 points overall. The elapsed runtime, including the input, formatting, preprocessing, identification, and extraction of damage events, the application of the FFT, and the extraction of points, was in the range of approximately 14 s on average for completion and plotting. For flexural testing, an increase in the length of the input was observed, i.e., lengths of 425–475 million data points (425–475 s). However, as flexural testing is far less active in terms of the AE activity generated, an average of 3000 damage events were observed per test, which, combined, yielded a total of 15,000 points. Despite this, an average elapsed runtime of approximately 18 s was observed per test dataset. The dominance of the relationship between the processing time required and the length of the input data was evident, rather than the number of identified events. The increase in time observed between the two input types generally demonstrated the good scalability of the processes described.

#### 3.2.2. Wavelet Transform

WTs are used to decompose the input signal into different levels, such that the spectral components can be directly assessed. The means by which this is performed is specified through the wavelet type selected for the decomposition itself, with the number of levels being alterable and operator-specific. The specifics of this given operation have been well explored and developed by Qi, considering both the theoretical and physical implications of their application to AE information of this type [9].

To assess the spectral content of an input AE event, as discussed by Barile et al., the in-built functionality of MATLAB applications designed for WTs, specifically the wavelet signal analyser and signal multiresolution analyser, are sufficient for providing the spectral energy distribution of a given input signal. The application of the DWT is influenced by the wavelet type selected, influencing the information ultimately extracted from the input signal [9,10]. As the aim here was for these parameters to serve as a source of comparison using a limited sample to reflect previous related work, a full comprehensive review of the best-suited wavelet type and level of decomposition was outside of the scope of the present study. Through the application of the DWT in this manner, the specific control of the frequency ranges contained within each level of a given wavelet decomposition was achieved; hence, a direct overlap between this and the established ranges within Section 3.1 was not observed. However, through the selection of the correct wavelet type, Haar or Daubechies, and level of good overlap could be achieved. A correlation between the information presented on the frequency spectra content through the use of the WT and Fourier transform could be seen. For these reasons, as well as the more accurate control of the frequency band selection, at least in the method chosen, the empirical wavelet transform (EWT) was selected for use and served as a point of comparison. Within the EWT, the decomposition of the signal is performed based on segmentation detected through the frequency spectra of a given input signal, upon which the signal division is based [30]. However, as this is performed sample by sample, it may give rise to variations in the assigned levels dependent on the frequency distribution of the input signal, as well as being, overall, a less efficient approach than a predetermined DWT due to the additional required processes.

Figure 11 demonstrates the processing associated with the completion of the EWT in this manner, with Figure 11a showing an example input damage waveform, and Figure 11b showing the wavelet components produced as a result of the performance of the EWT. Each of these levels was limited between frequency bands. In turn, like in the cases where the DWT was applied, this allowed for the evaluation of the spectral energy of a given input signal and the distribution of energy amongst the frequency range. From the above example, it was found that over 75% of the spectral energy was within the range associated with matrix cracking.

## 4. Discussion

### 4.1. Method Comparison

The two methods explored in the present study investigate the frequency components of a given input signal in a different manner. The FT-based approach is a direct expansion of the existing peak frequency assessment approach in order to include events beyond the dominant damage mode observed. The WT method based on EWT explores the energy contained within a specified frequency range extracted from a signal by its decomposition. From the examples for the techniques demonstrated in Section 3.2.1 and Section 3.2.2, a direct comparison between the proposed multi-variant method, and the WT method can be made. From the identified damage events across the datasets, the correlation regarding the types of damage present and their ratio relative to one another can be established. What clearly differentiates the two approaches is the manner in which this information is interpreted.

The FFT-based approach gives a direct peak associated with a detected frequency within the damage event, whose magnitude can be evaluated directly. This allows for tracking the progression of a specific damage mode through its initiation and propagation based on the value reflected here, as opposed to a generalised level of a signal which can be attributed to that range. However, it is not certain that the same specific instance of damage can be tracked directly within this environment with the monitoring setup described in Section 2.3. Yet, it can provide earlier indications of the occurrence of various damage modes simultaneously, which when considering their interlinked nature in some cases, such as matrix cracking and delamination, may prove particularly useful in the analysis of the health of a structure, as well as the expected lifetime. For example, an early detection of delamination, through the detection of its initiation in this way, may prevent the risks associated with such a damage site present in a structure; this is further explored within the subsequent section.

For the WT method, as explored by Qi and Barile et al., which is further investigated here, evaluation of the spectral energy can be used to assess the health of the monitored structure/component during monitoring. Evaluation of the spectral energy allows for this, as it provides a measure of the energy of a given signal contained within the specific highlighted frequency range for a given wavelet. Applying the work completed within Section 3.1, with the accurate designation of frequency ranges that apply, allows for the general assessment of the proportion of the energy contained within this captured signal, which can be attributed to a particular damage mode, as demonstrated in the example in the above section. This provides a different type of information regarding the material, in the sense that it is not known what specific events occur at a given time. Instead, the level of a specific mode is known relative to the others. From the perspective of SHM or CM, this may be sufficient to assess whether the damage that has occurred to a specific structure or component is of a type or level that is deemed acceptable, for example, matrix cracking dominating rather than a more severe type of damage mode, such as delamination. For ease, this can be seen to be a simpler approach than the alternative presented above. The main limitation of this technique is that it does not focus on a specific event and may adversely influence the ability to track or monitor a specific damage mode.

### 4.2. Tracking Damage Propagation

As discussed earlier, the proposed methodology has the ability to detect and quantify the damage events identified. Thus, it can be used as an early-warning system for the detection and quantification of initiation of a specific damage mode, as well as to evaluate its propagation in service. To best achieve this, an individual damage mode was selected for consideration and, hence, its associated frequency range. In this case, delamination was selected. Moreover, to allow for further interrogation of the events being displayed, a third axis was introduced within the plots, displaying the magnitude of each of the events, allowing for inter-level comparison, as well as the study of the progression of damage in terms of its intensity. To demonstrate the benefit of such an approach, the first instance of delamination observed within each of the levels was highlighted, providing its time, frequency, and magnitude information, displayed as *X*, *Y*, and *Z*, respectively as shown in Figure 12.

When the AE frequency content is analysed in this way, the primary benefit becomes apparent, as demonstrated by the occurrence of an event indicative of delamination being detected sooner, at ~42 s for level 2, indicative of a delamination event, compared with ~70 s for level 1, equivalent to peak frequency assessment. What is also gained through the introduction of the third “magnitude” axis is the overall higher magnitude seen for this event type when comparing the lower levels with level 1 (peak). From this, it can be inferred that in the assessment using peak frequency alone, these relatively high magnitude delamination events would be masked by the simultaneous occurrence of a damage mode of a higher signal intensity. As matrix cracking generally occurs at lower loading levels, crack growth here would generally be of a greater intensity than the initiation of another damage mode within the structure, hence its masking.

## 5. Conclusions

Damage in FRP materials and structures can initiate and propagate through complex mechanisms. This means damage characterisation and the accurate assessment of the severity of damage detected, particularly when using passive monitoring methods, such as AE, poses significant challenges. This study successfully developed a novel multi-variant frequency assessment methodology for real-time damage characterisation in FRP materials using AE techniques. The approach advances beyond conventional peak frequency analysis by implementing fast Fourier transform-based evaluation of multiple frequency peaks within AE signals, enabling the simultaneous detection of five distinct damage mechanisms: matrix cracking (100–200 kHz), delamination (205–265 kHz), debonding (270–320 kHz), fibre fracture (330–385 kHz), and fibre pullout (395–490 kHz). The methodology demonstrated significant improvements in early damage detection. Validation through k-means and DBSCAN clustering algorithms confirmed the accuracy of frequency band classification, while comparative analysis with WT methods highlighted the superior control and direct damage-tracking capabilities of the FFT-based approach. The technique enables the continuous, real-time monitoring of large-scale FRP structures without system shutdown, making it particularly valuable for remote installations, such as wind turbines, where conventional inspection methods are challenging and costly. The presented approach can contribute towards condition-based and predictive maintenance scheduling based on actual structural conditions rather than predetermined intervals, significantly reducing operational downtime and maintenance costs while enhancing structural reliability and safety in critical infrastructure applications, such as wind turbine blades.

## Figures and Tables

**Figure 1 sensors-25-03795-f001:**
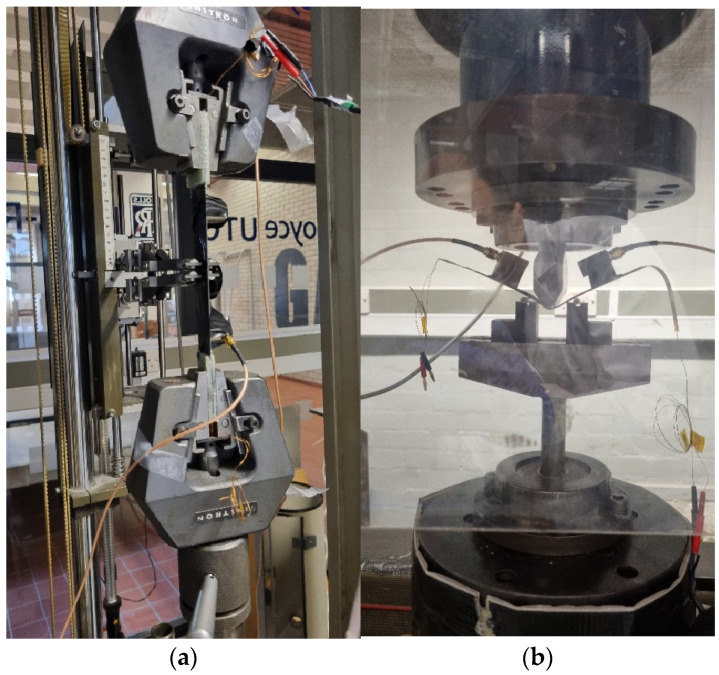
Mechanical testing configurations for (**a**) tensile and (**b**) 3-point bending (flexural).

**Figure 2 sensors-25-03795-f002:**
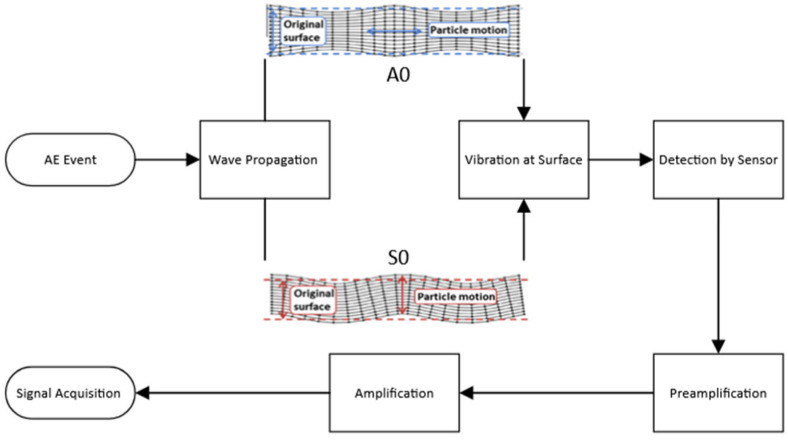
Simplified schematic showing AE acquisition architecture employed, adapted from [16].

**Figure 3 sensors-25-03795-f003:**
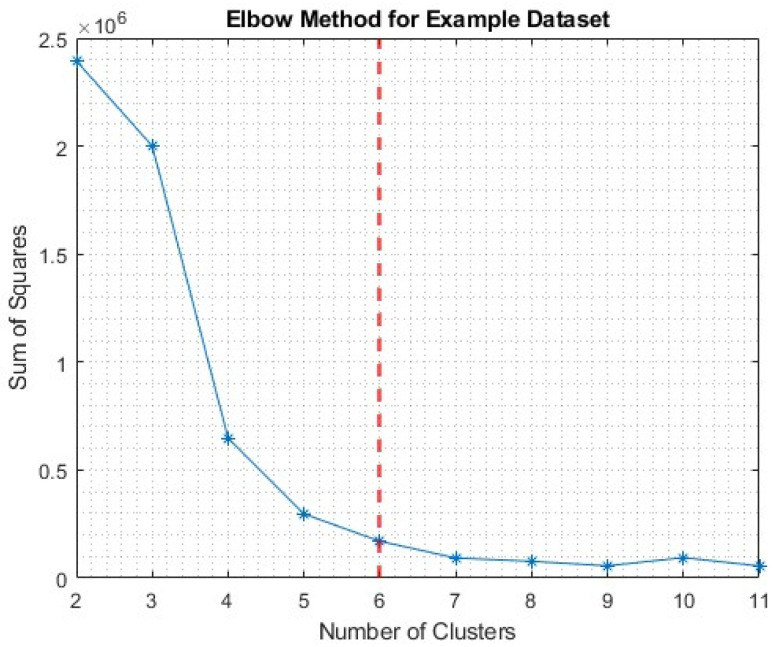
Elbow method assessment for tensile example dataset.

**Figure 4 sensors-25-03795-f004:**
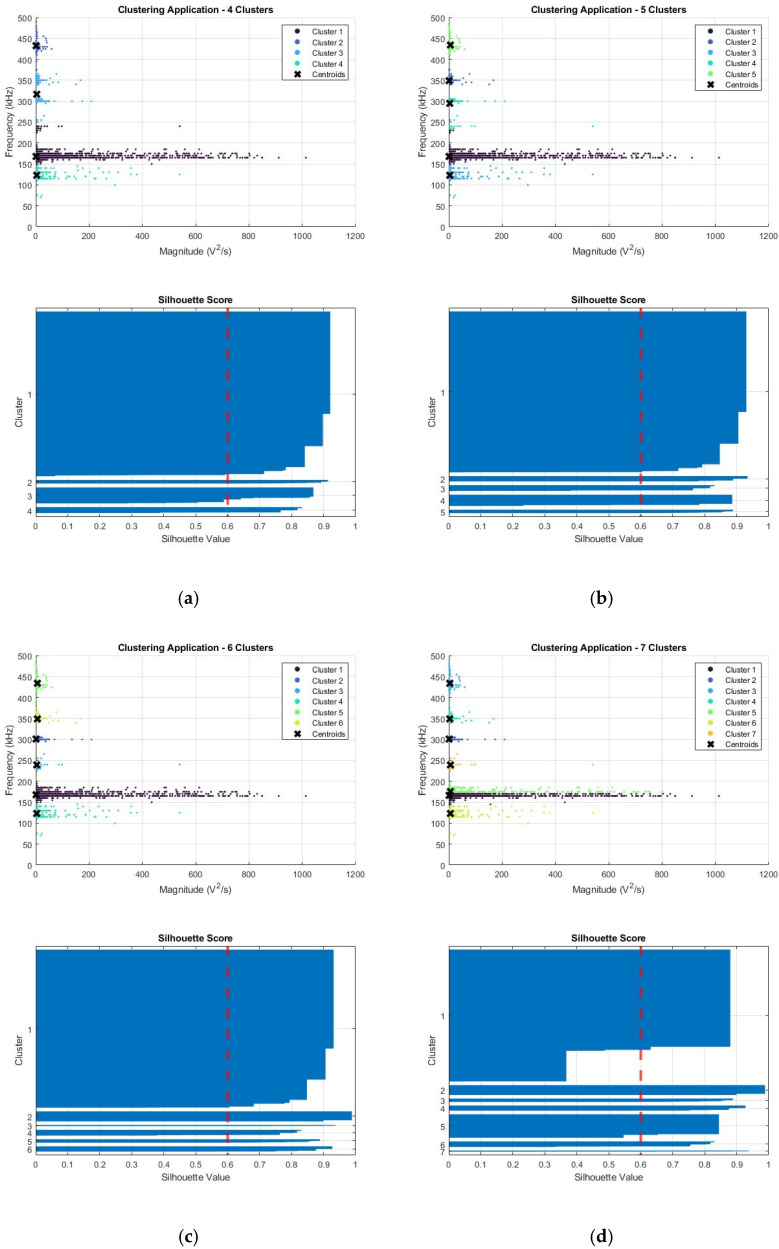
Application of k-means and respective silhouette score for (**a**) 4, (**b**) 5, (**c**) 6, and (**d**) 7 clusters on example tensile dataset.

**Figure 5 sensors-25-03795-f005:**
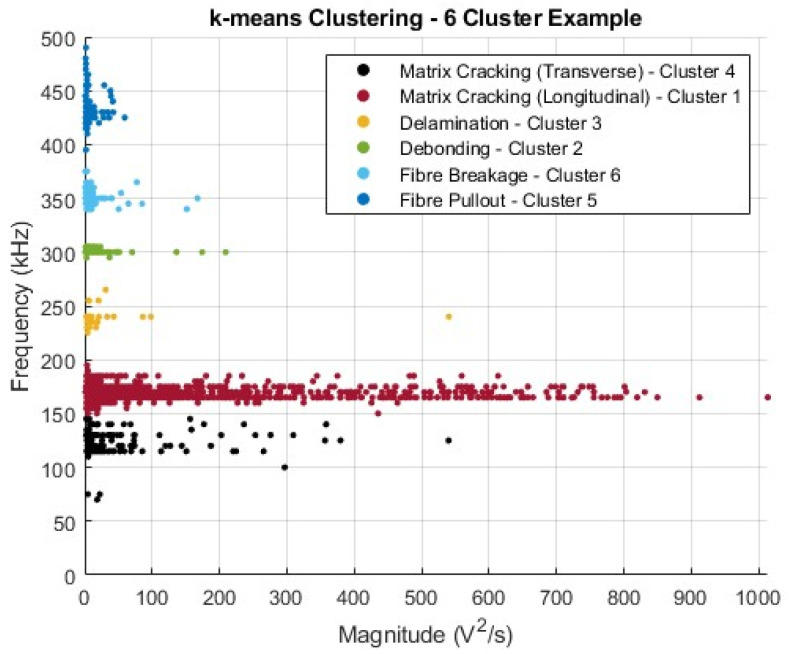
Six-cluster k-means example with assigned damage modes shown.

**Figure 6 sensors-25-03795-f006:**
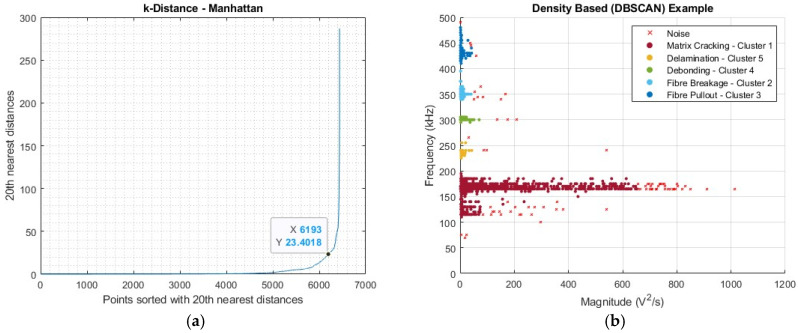
Application of DBSCAN: (**a**) k-distance measure assessment and (**b**) application of clustering technique.

**Figure 7 sensors-25-03795-f007:**
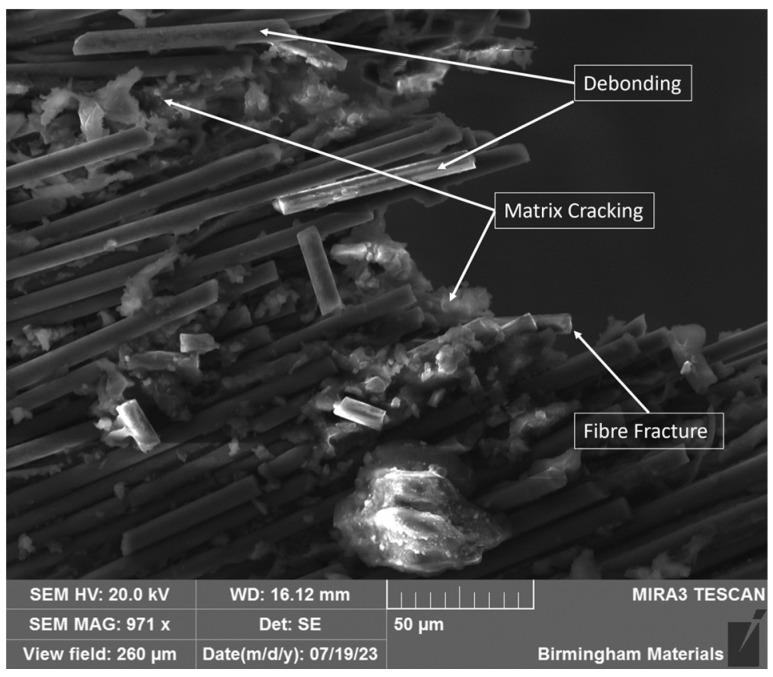
SEM image of fracture site for selected tensile sample.

**Figure 8 sensors-25-03795-f008:**
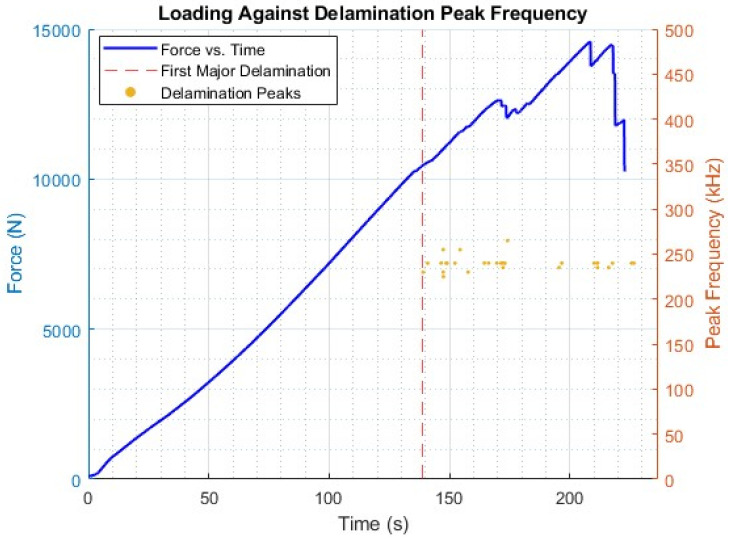
Loading trace for tensile example against delamination frequency peaks.

**Figure 9 sensors-25-03795-f009:**
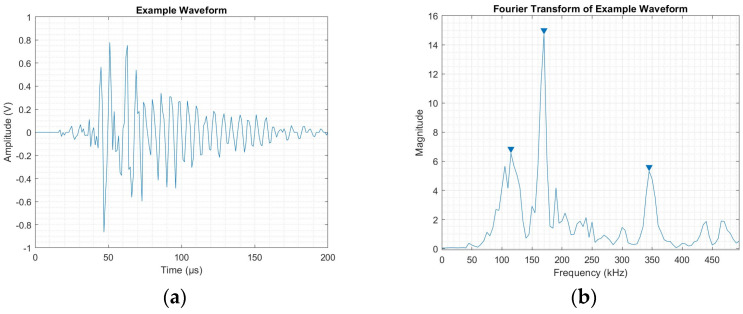
(**a**) Example AE waveform and (**b**) performed selection parameters on FFT of reference waveform.

**Figure 10 sensors-25-03795-f010:**
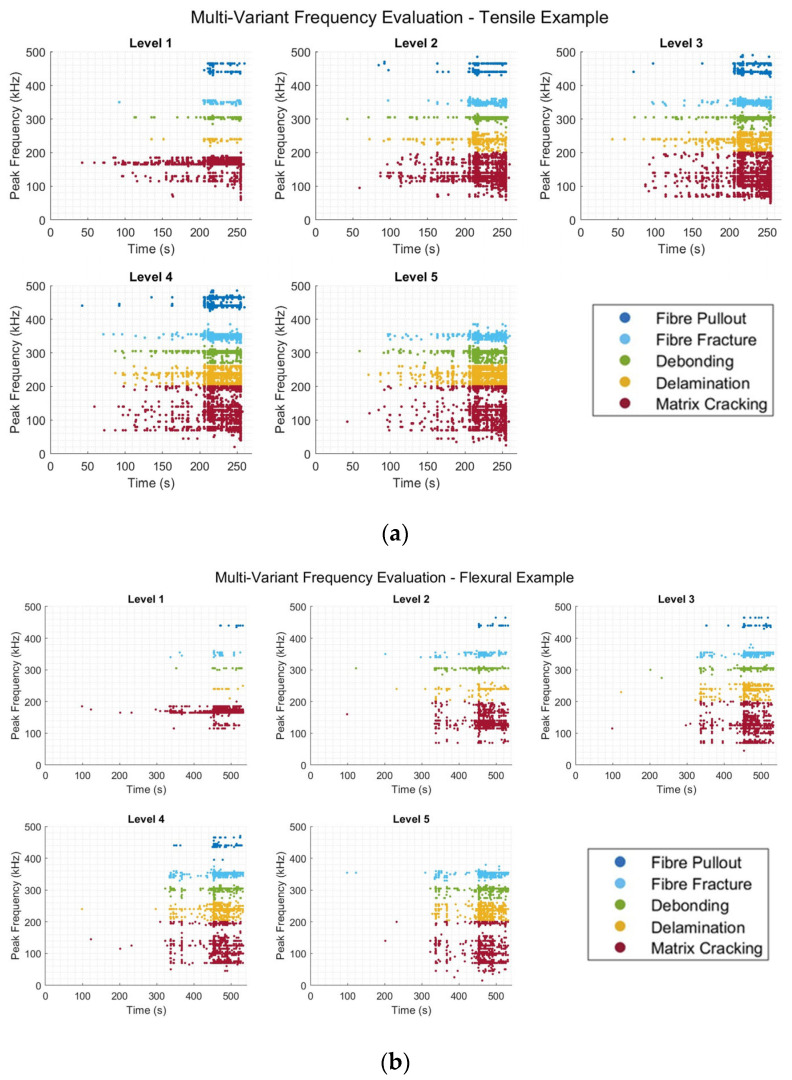
Multi-variant analysis showing individual intensity levels: visual intensity levels for (**a**) a tensile and (**b**) a flexural test sample.

**Figure 11 sensors-25-03795-f011:**
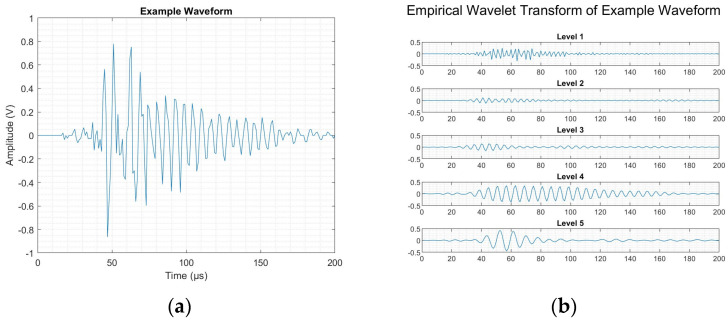
Damage event: (**a**) example waveform and (**b**) its decomposition using EWT.

**Figure 12 sensors-25-03795-f012:**
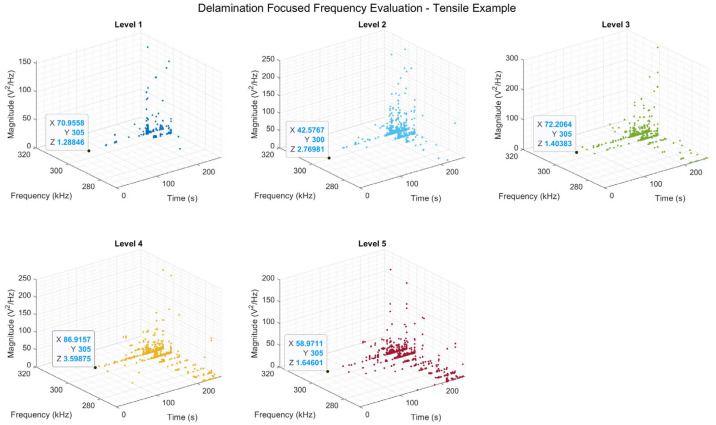
Delamination-focused damage assessment for a tensile sample. Different colours have been used to differentiate between the energy levels.

**Table 1 sensors-25-03795-t001:** Extracted frequency ranges and respective damage modes for CFRP using k-means.

Damage Mode	Lower Limit (kHz)	Upper Limit (kHz)	Colour
Matrix cracking	100	200	Red
Delamination	205	265	Orange
Debonding	270	320	Green
Fibre fracture	330	385	Cyan
Fibre pullout	395	490	Blue

**Table 2 sensors-25-03795-t002:** Frequency ranges with respective damage modes reported in various other studies.

Damage Mode	Literature Value (kHz)		
	Bussiba et al. [3]	Gutkin et al. [19]	Wirtz et al. [29]
Matrix cracking	140	0–50	95
Delamination	-	50–150	45
Debonding	300	200–300	245
Fibre fracture	405	400–500	300
Fibre pullout	-	500–600	-

## Data Availability

The data that support the findings of this study are available from the authors upon reasonable request. Please contact the corresponding author for any data access requests.

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
