# Peer review of "Multi-Variant Damage Assessment in Composite Materials Using Acoustic Emission"

_sensors, 2025, doi:10.3390/s25123795_

Round 1
Reviewer 1 Report
Comments and Suggestions for Authors
This article investigated the application of frequency-based multi-variant analysis of acoustic emission signals to accurately identify and quantify multiple damage modes in fiber-reinforced polymer composites during mechanical testing. The paper is well-written and organized, At the same time, the results obtained were correctly validated by comparison with data obtained from the pertinent literature. Considering this important positive aspect, the present paper should be considered for publication, but the Authors must also consider the following critical elements in the revised version.
1- This paper should be edited grammatically. Please check the whole manuscript for typos and punctuation mistakes, such as Definite and Indefinite Articles (a, an, the).
2- Some important findings can be presented in the Abstract.
3- Explain Clearly Figure 2. (please clarify the labeling of axes, the meaning of each cluster (color/region), how the number of clusters was determined, and how each frequency band corresponds to specific damage modes.)
4- Explain Clearly Figure 5. (please define what each “intensity level” represents, clarify the labeling of all axes, explain the color coding or data point distribution, and state whether the data shown are from a single sample or aggregated from multiple tests.)
5- How was the acoustic emission-based damage identification validated against ground truth? Were complementary methods such as microscopy or ultrasonic testing used after mechanical tests to confirm the detected damage modes?
6- Could the authors elaborate on how the number of clusters was determined for the k-means clustering? Was any objective method (e.g., the Elbow Method, Silhouette Score) used to justify the choice of six clusters? Additionally, please discuss the sensitivity of the clustering results to the number of clusters selected.
7- The current references are appropriate. I would suggest incorporating additional references to strengthen the analysis and broaden the context of the study. Here are some recommended sources that could enhance the depth and relevance of the article’s findings.
- https://doi.org/10.36074/logos-18.10.2024.056
- https://doi.org/10.1016/j.ijft.2023.100409
Therefore, considering the previous observations, the Reviewer believes this paper should be accepted for Major revisions.

Author Response
Dear Reviewer 1,
We would like to express our gratitude for your useful criticism and constructive comments which have helped us to improve and enhance our paper further. Your criticism and comments have been addressed in the revised version of our article submitted herewith for further consideration. Below we respond to your comments.
Reviewer 1: This article investigated the application of frequency-based multi-variant analysis of acoustic emission signals to accurately identify and quantify multiple damage modes in fibre-reinforced polymer composites during mechanical testing. The paper is well-written and organized, At the same time, the results obtained were correctly validated by comparison with data obtained from the pertinent literature. Considering this important positive aspect, the present paper should be considered for publication, but the Authors must also consider the following critical elements in the revised version.
Authors response: Thank you for your positive comments. We have addressed the following critical elements in the attached revised version.
Reviewer 1- This paper should be edited grammatically. Please check the whole manuscript for typos and punctuation mistakes, such as Definite and Indefinite Articles (a, an, the).
Authors response: The manuscript has been checked grammatically by the authors and other colleagues, in order to minimise any discrepancies.
Reviewer 1: 2- Some important findings can be presented in the Abstract.
Authors response: Thank you for your comment, this has been updated within the reviewed manuscript. The abstract now includes the identified damage modes and their respective frequency ranges, the ability of the proposed technique, as well as the corresponding uses for such a technique.
Reviewer 1: 3- Explain Clearly Figure 2. (please clarify the labeling of axes, the meaning of each cluster (color/region), how the number of clusters was determined, and how each frequency band corresponds to specific damage modes.)
Authors response: Thank you for your comment, on review this section appeared to be lacking detail for those wishing to both understand and repeat any work completed here. The main comments have been addressed within the text, however, to summarise silhouette score was used to confirm correct cluster number assignment and the final link between clusters and relevant damage mode were made by the operator, yet sorting these numerically in terms of frequency helped in achieving this.
Reviewer 1: 4- Explain Clearly Figure 5. (please define what each “intensity level” represents, clarify the labeling of all axes, explain the color coding or data point distribution, and state whether the data shown are from a single sample or aggregated from multiple tests.)
Authors response: Thank you for your comment, this highlighted possible confusion and lack of clarity which would be associated with the figure. To address this subtitles for each of these have been introduced to better highlight their content and description of this with reference to the reasoning and origin for this within the text following.
Reviewer 1: 5- How was the acoustic emission-based damage identification validated against ground truth? Were complementary methods such as microscopy or ultrasonic testing used after mechanical tests to confirm the detected damage modes?
Authors response: Herewith, as testing was completed until failure for each of the test coupons, UT testing would not be possible to use as the sample is effectively destroyed at the end of the test. Its very small thickness is also an additional limitation. Themoregraphy could have been used if the test was not done to failure. In the present study we have used microscopy instead, both optical and SEM, to confirm the presence of the damage modes mentioned and the severity of these.
Reviewer 1: 6- Could the authors elaborate on how the number of clusters was determined for the k-means clustering? Was any objective method (e.g., the Elbow Method, Silhouette Score) used to justify the choice of six clusters? Additionally, please discuss the sensitivity of the clustering results to the number of clusters selected.
Authors response: An explanation of measures taken in order to improve the accuracy and account for the limitations of the k-means technique. Be that the elimination of outliers without effecting the quality and accuracy of the data, as well as confirmation of the number of clusters for specific datasets through use of supplementary techniques such as the silhouette score to ensure that the correct parameters are selected prior to its implementation.
Reviewer 1: 7- The current references are appropriate. I would suggest incorporating additional references to strengthen the analysis and broaden the context of the study. Here are some recommended sources that could enhance the depth and relevance of the article’s findings.
- https://doi.org/10.36074/logos-18.10.2024.056
- https://doi.org/10.1016/j.ijft.2023.100409
Authors response: Thank you for the useful input. On review of the papers mentioned in the comment, we are of the opinion that the first suggestion does bear some crossover in terms of content, although it is brief in its considerations, primarily for the use of k-means for the establishment of clusters pertaining to specific damage modes. This has been included as a reference in the relevant section. The latter of the two, whilst using some acoustic based techniques, they investigate Rayleigh waves and behaviours of acoustics in thick materials, where these waves are observed, with no real semblance of the material or AE technique applied within the paper.
Reviewer 1: Therefore, considering the previous observations, the Reviewer believes this paper should be accepted for Major revisions.
Authors response: Thank you for the comment and constructive observations made. We have carried out the necessary major revisions which we hope have addressed the points you have raised.
On behalf of the authors,
Yours sincerely,
Professor Mayorkinos Papaelias
Reviewer 2 Report
Comments and Suggestions for Authors
Multi-Variant Damage Assessment in Composite Materials Using Acoustic Emission is presented in this manuscript, which is well structured and of innovation. Minro revision are necessary before publication. Comments are below.
- In the description of materials, only basic information such as manufacturing process, dimensions, and layup structure of the test specimens is provided. Specific models and performance parameters of the resin and fibers used, like the elastic modulus of the resin and the tensile strength of the fibers, are lacking. This may affect the accurate understanding of experimental materials and the reproducibility of the research by other scholars. Authors should add these key material parameters.
- In the mechanical testing, although the equipment, loading speed, and other conditions for tensile and flexural tests are introduced, the test environmental conditions, such as temperature and humidity, are not mentioned. These environmental factors may influence the performance and damage behavior of FRP materials, and thus affect the characteristics of acoustic emission signals. Relevant information on test environmental conditions should be supplemented to make the description of the experimental procedure more complete.
- When using k - means clustering to determine the frequency ranges of damage modes, the paper mentions that operator input, such as the number of clusters, is required, and the k - means clustering algorithm is run multiple times to ensure the accuracy of clustering. However, there is a lack of detailed evaluation methods and judgment criteria for determining the consistency of multiple clustering results and the rationality of the final clustering result. Authors should add relevant content, for example, what statistical indicators are used to measure the stability of clustering results, to enhance the reliability and scientific nature of the research results.
- Some of the annotations in the figures and tables in the paper are not clear enough. For example, the units of the coordinate axes in Figure 2 are not clearly marked, and the meanings represented by different colors in Figure 5 are not described in detail Authors should carefully check all figures and tables to ensure that the annotations are complete and clear.
Author Response
Dear Reviewer 2,
We would like to express our gratitude for your useful criticism and constructive comments which have helped us to improve and enhance our paper further. Your criticism and comments have been addressed in the revised version of our article submitted herewith for further consideration. Below we respond to your comments.
Reviewer 2: Multi-Variant Damage Assessment in Composite Materials Using Acoustic Emission is presented in this manuscript, which is well structured and of innovation. Minor revision are necessary before publication. Comments are below.
Authors response: Thank you for your positive assessment. We have addressed the following comments you have made in the submitted revised version.
Reviewer 2- 1. In the description of materials, only basic information such as manufacturing process, dimensions, and layup structure of the test specimens is provided. Specific models and performance parameters of the resin and fibers used, like the elastic modulus of the resin and the tensile strength of the fibers, are lacking. This may affect the accurate understanding of experimental materials and the reproducibility of the research by other scholars. Authors should add these key material parameters.
Authors response: Thank you for your comment, for the purpose of repeatability, providing this information for the reader is of great importance. Therefore, a value for the Young’s modulus has been provided within the text, which was provided by the supplier of the test coupons.
Reviewer 2: 2- In the mechanical testing, although the equipment, loading speed, and other conditions for tensile and flexural tests are introduced, the test environmental conditions, such as temperature and humidity, are not mentioned. These environmental factors may influence the performance and damage behaviour of FRP materials, and thus affect the characteristics of acoustic emission signals. Relevant information on test environmental conditions should be supplemented to make the description of the experimental procedure more complete.
Authors response: For the testing of the CFRP coupons, this was completed within an environmentally controlled facility, such that the temperature and humidity were constant throughout. A comment has been included within the manuscript to clarify the stability of these conditions throughout the testing.
Reviewer 2: 3- When using k - means clustering to determine the frequency ranges of damage modes, the paper mentions that operator input, such as the number of clusters, is required, and the k - means clustering algorithm is run multiple times to ensure the accuracy of clustering. However, there is a lack of detailed evaluation methods and judgment criteria for determining the consistency of multiple clustering results and the rationality of the final clustering result. Authors should add relevant content, for example, what statistical indicators are used to measure the stability of clustering results, to enhance the reliability and scientific nature of the research results.
Authors response: An explanation of measures taken in order to improve the accuracy and account for the limitations of the k-means technique. Be that the elimination of outliers without effecting the quality and accuracy of the data, as well as confirmation of the number of clusters for specific datasets through use of supplementary techniques such as the silhouette score to ensure that the correct parameters are selected prior to its implementation.
Reviewer 2: 4- Some of the annotations in the figures and tables in the paper are not clear enough. For example, the units of the coordinate axes in Figure 2 are not clearly marked, and the meanings represented by different colors in Figure 5 are not described in detail Authors should carefully check all figures and tables to ensure that the annotations are complete and clear.
Authors response: Thank you for your comment, this highlighted possible confusion and lack of clarity which would be associated with the figure. To address units have been introduced here as well as, subtitles for each of these have been introduced to better highlight their content and description of this with reference to the reasoning and origin for this within the text following.
Reviewer 2: Discusses about developing and validating a new method for assessing damage in fibre-reinforced polymer (FRP) materials using Acoustic Emission (AE) signals, with a focus on improving how different types of damage are detected and classified during mechanical loading (like tensile and flexural tests). In this regard, the Authors propose a multi-variant, frequency-based analysis where more than one frequency peak is considered per event, instead of just the dominant one.
The paper is overall interesting, but it is hampered by several limitations, both conceptually and in its editing. Nevertheless, it could be further reconsidered for potential acceptance, if all the following points are properly addressed in the revised version.
Authors response: We are glad that you have found the paper interesting. We hope we have addressed your concerns highlighted in your comments in the revised version.
On behalf of the authors.
Yours sincerely,
Professor Mayorkinos Papaelias
Reviewer 3 Report
Comments and Suggestions for Authors
The research paper
‘Multi-Variant Damage Assessment in Composite Materials Using Acoustic Emission‘
By Gee et al
Discusses about developing and validating a new method for assessing damage in fibre-reinforced polymer (FRP) materials using Acoustic Emission (AE) signals, with a focus on improving how different types of damage are detected and classified during mechanical loading (like tensile and flexural tests). In this regard, the Authors propose a multi-variant, frequency-based analysis where more than one frequency peak is considered per event, instead of just the dominant one.
The paper is overall interesting, but it is hampered by several limitations, both conceptually and in its editing. Nevertheless, it could be further reconsidered for potential acceptance, if all the following points are properly addressed in the revised version.
- The discussion lightly mentions limitations (e.g., sensor bandwidth affecting low-frequency data) but a clearer and more structured "Limitations" subsection would strengthen the article’s structure.
- More specific details on the k-means clustering process are needed, especially on how the number of clusters was decided and validated. The current explanation is a bit vague
- k-means clustering is widely known to partition the search space into k regions of similar, as it assumes all clusters have similar variances. This causes strong issues when the dataset is not divided in populations of equal size.
- Sections 2.1 and 2.2: if available, it would be useful to add pictures of the material and experimental setup
- The Section explaining the Wavelet Transform (WT) is underdeveloped relative to the Fourier-based method. As it is mentioned, it should be better expanded to match the length and level of detail of its counterpart.
- Although a comparison is attempted, it remains very general (e.g., "general ordering [..] aligns with literature"). It would benefit from deeper numerical comparison with at least one or two established studies
- Related to the above, the literature review concerning FRP materials is limited. The Authors should expand it, referencing to works such as Treed gaussian process for manufacturing imperfection identification of pultruded GFRP thin-walled profile, Recursive partitioning and Gaussian Process Regression for the detection and localization of damages in pultruded Glass Fiber Reinforced Polymer material, and similar ones.
- Table captions, figure captions (e.g., Figure 1, Figure 2), and inline references need refinement for clarity and professional formatting.
- The Flowchart in Figure 1 may be graphically improved
- Title of Figure 2, correct ‘Frequecny’.
- Figure 5, assign titles to each subplots
- The conclusions should be expanded and deepened.
The English is overall acceptable but should be improved. Some sentences are long and would benefit from minor editing for clarity and conciseness (e.g., some in Sections 1.3 and 3.2).
Author Response
Dear Reviewer 3,
We would like to express our gratitude for your useful criticism and constructive comments which have helped us to improve and enhance our paper further. Your criticism and comments have been addressed in the revised version of our article submitted herewith for further consideration. Below we respond to your comments.
Reviewer 3: 1 - The discussion lightly mentions limitations (e.g., sensor bandwidth affecting low-frequency data) but a clearer and more structured "Limitations" subsection would strengthen the article’s structure.
Authors response: Thank you for the comment. The limitations have been elaborated further.
Reviewer 3: 2- More specific details on the k-means clustering process are needed, especially on how the number of clusters was decided and validated. The current explanation is a bit vague
Authors response: Thank you for your comment, initially this was intended as a supplementary technique, however, due to the mention of this by multiple reviewers this has new been addressed within the manuscript. This addresses the pre-processing of the dataset to improve the sensitivity of the technique, additionally the selection of number of clusters has been clarified, including information on the techniques applied (silhouette score) for the confirmation for each individual dataset.
Reviewer 3: 3- k-means clustering is widely known to partition the search space into k regions of similar, as it assumes all clusters have similar variances. This causes strong issues when the dataset is not divided in populations of equal size.
Authors response: Thank you for your comment, this is appreciated, however, the nature of the data does not produce clusters or a spread of data which reflects this. Priority has been placed on the accurate representation of the data collected within the work completed here, yet this would further rationalise the discrepancy seen in the accuracy of the clustering.
Reviewer 3: 4- Sections 2.1 and 2.2: if available, it would be useful to add pictures of the material and experimental setup
Authors response: Thank you for your comment, for the purpose of repeatability, inclusion of this important, hence a figure containing images of both the testing setups used has now been included.
Reviewer 3: 5-The Section explaining the Wavelet Transform (WT) is underdeveloped relative to the Fourier-based method. As it is mentioned, it should be better expanded to match the length and level of detail of its counterpart.
Authors response: We appreciate your comment. However, this is not the intention of the paper or the work completed within. The primary focus of the work is upon the work relying on the Fourier transform and the identification of specific events using this. The supplementary section on Wavelet transforms was included to provide a direct reference point to work found within the literature regarding the full appreciation and evaluation of the frequency spectrum. In this respect the authors believe that the information and the work completed using EWT is sufficient in this regard. It highlights both how this technique operates and the information which can be ascertained in this manner, the spectral energy of a frequency range. The work produced results to the same level of detail as the referenced sources and includes information regarding how this was achieved. As no novel work is completed here further expansion is not believed to be necessary.
Reviewer 3: 6- Although a comparison is attempted, it remains very general (e.g., "general ordering [..] aligns with literature"). It would benefit from deeper numerical comparison with at least one or two established studies
Authors response: Thank you for comment, reference was made to sources, however, no explicit value were provided for these. To rectify this an additional table including example literature values has been included within the updated manuscript.
Reviewer 3: 7- Related to the above, the literature review concerning FRP materials is limited. The Authors should expand it, referencing to works such as Treed gaussian process for manufacturing imperfection identification of pultruded GFRP thin-walled profile, Recursive partitioning and Gaussian Process Regression for the detection and localization of damages in pultruded Glass Fiber Reinforced Polymer material, and similar ones.
Authors response: On review of the suggested paper it is the opinion of the author that this work does not share enough similarity with the work being presented. Chiefly two separate techniques are used for monitoring, in the study acoustic emission is used to passively monitor damage initiation within a structure or component during its loading. However, the suggested literature is based upon use of vibrational sensors to detect deviation in frequency from a direct excitation source due to the presence of defect within the structure already, which is interesting, but is more similar in nature to methods such as ultrasonic inspection or impact echo testing.
Reviewer 3: 8- Table captions, figure captions (e.g., Figure 1, Figure 2), and inline references need refinement for clarity and professional formatting.
Authors response: Thank you for the comment. The table and figure captions have been revised along with references and reformatted.
Reviewer 3: 9- The Flowchart in Figure 1 may be graphically improved
Authors response: The schematic is effective in demonstrating the procedure included and required for the acquisition process. To make it more accessible, the wave types have been included to show the motion included within the material for the A0 and S0 mode.
Reviewer 3: 10- Title of Figure 2, correct ‘Frequecny’.
Authors response: Thank you for the comment and raising this, this has now been addressed within the manuscript
Reviewer 3: 11- Figure 5, assign titles to each subplots
Authors response: Thank you for your comment, for clarity this would be beneficial, hence has been introduced within the manuscript.
Reviewer 3: 12- The conclusions should be expanded and deepened.
Authors response: This section has been expanded in order to deeper explore the findings, their rationalisation and the implications of these, providing a better overall summary of the study.
On behalf of the authors.
Yours sincerely,
Professor Mayorkinos Papaelias
Round 2
Reviewer 1 Report
Comments and Suggestions for Authors
Dear Authors,
Thank you for submitting the revised manuscript and addressing the initial reviewer comments. However, several issues remain that require further attention.
Please address the following specific concerns in your revised manuscript. Detailed responses to each point, along with the revised manuscript, are expected. The key issues are as follows:
1. Several grammatical and punctuation errors persist in the revised manuscript (e.g., repetitive phrasing on page 7: “k-means is sensitive to outliers within the dataset as it is highly sensitive to them” and inconsistent use of articles like “the” and “a” on page 5).
2. The axes labeling in the figure showing “Frequency with respect to magnitude” (presented on page 8) is not clearly defined (e.g., units and scales are missing). Please specify the units and scales for both axes, explain the meaning of each color or region representing different clusters, and provide a detailed description of how these clusters were visually or numerically distinguished in the figure.
3. The axes in Figure 5 lack clear labeling (e.g., what do the x- and y-axes represent?). Additionally, it is unclear whether the data points are from a single sample or aggregate from multiple tests (9 tensile and 7 flexural samples). Please label all axes with appropriate units, define the data source (single sample or aggregated), and explain the color coding scheme in relation to the damage modes listed in Table 1.
4. You mentioned using optical and SEM microscopy to validate damage modes, but no supporting data (e.g., images or quantitative results) were provided. Please include representative microscopy images (optical and/or SEM) for at least three damage modes (e.g., matrix cracking, delamination, and fibre fracture) and correlate these findings with the acoustic emission frequency ranges reported in Table 1 to strengthen the validation.
5. The use of the Silhouette Score to determine the number of clusters is noted, but the Elbow Method was not mentioned, and a quantitative sensitivity analysis of the clustering results to the number of clusters (e.g., 5, 6, or 7 clusters) is missing. Please provide a detailed explanation of why six clusters were chosen, include results from both the Silhouette Score and Elbow Method, and present a quantitative assessment of how changing the number of clusters affects the assignment of damage modes.
Author Response
Dear Reviewer 1,
Thank you for the additional valuable constructive comments which have helped us to improve our manuscript. We have addressed the comments in the revised manuscript to enable its further consideration for publication.
Below please find our response to your comments.
- Several grammatical and punctuation errors persist in the revised manuscript (e.g., repetitive phrasing on page 7: “k-means is sensitive to outliers within the dataset as it is highly sensitive to them” and inconsistent use of articles like “the” and “a” on page 5).
Thank you for comment, care has been taken to check the manuscript to minimise the number of mistakes and introduce a consistent language style for the ease of the reader and quality of the paper.
2. The axes labeling in the figure showing “Frequency with respect to magnitude” (presented on page 8) is not clearly defined (e.g., units and scales are missing). Please specify the units and scales for both axes, explain the meaning of each color or region representing different clusters, and provide a detailed description of how these clusters were visually or numerically distinguished in the figure.
Thank you for your comment, this figure has been changed to another now, see comment 5, but these concerns have been addressed within this new figures, with measures taken for the ease of the reader.
- The axes in Figure 5 lack clear labeling (e.g., what do the x- and y-axes represent?). Additionally, it is unclear whether the data points are from a single sample or aggregate from multiple tests (9 tensile and 7 flexural samples). Please label all axes with appropriate units, define the data source (single sample or aggregated), and explain the color coding scheme in relation to the damage modes listed in Table 1.
Thank for your comment. These labels for the axes including units are on each of the individual subplots within the figure, now figure 10. A column has been introduced into table one which designates the colour used for representation for that specific damage mode throughout the remainder of the transcript. Additionally, comments have been added within the manuscript specifying the origin of the data used in each of the plots shown.
- You mentioned using optical and SEM microscopy to validate damage modes, but no supporting data (e.g., images or quantitative results) were provided. Please include representative microscopy images (optical and/or SEM) for at least three damage modes (e.g., matrix cracking, delamination, and fibre fracture) and correlate these findings with the acoustic emission frequency ranges reported in Table 1 to strengthen the validation.
This inspection was completed after testing of a given sample was completed. Therefore, it is possible to use this to confirm the occurrence of given damage modes within the testing of the sample coupon, however, cannot be used to correlate damage and frequencies as it is not known when these have occurred. Completion of these tests at different stages within the testing cycle would serve to improve the accuracy and validity of the findings presented here. These images have been included nonetheless as well as an alternative method involving using the loading trace to confirm delamination and the matching of this to observed AE data to validate this assigned region
- The use of the Silhouette Score to determine the number of clusters is noted, but the Elbow Method was not mentioned, and a quantitative sensitivity analysis of the clustering results to the number of clusters (e.g., 5, 6, or 7 clusters) is missing. Please provide a detailed explanation of why six clusters were chosen, include results from both the Silhouette Score and Elbow Method, and present a quantitative assessment of how changing the number of clusters affects the assignment of damage modes.
Thank you for comment regarding this, multiple measures have been taken to improve the validity of the results gathered in this section. The elbow method has now been included, as well as both visual and numerical evaluation for 4, 5, 6 and 7 clusters using the silhouette score. Additionally, an alternative form of clustering has been included, DBSCAN, which has automated designation of clusters and is fully unsupervised in this manner, in order to address some issues with k-means as well as serve to validate the results of this. These measures were completed for all samples, although only the one tensile sample is shown here as it is sufficient to demonstrate the discussed techniques.
On behalf of the authors,
Professor Mayorkinos Papaelias
Reviewer 3 Report
Comments and Suggestions for Authors
This Reviewer appreciates the Authors’ efforts to improve the manuscript and their engagement with the review comments.
Several revisions have addressed earlier concerns to some extent, particularly in areas like figure clarity, limitations, and the addition of comparative tables. However, this Reviewer believes a further round of major revisions is necessary, as many points remain to be addressed:
-
Limitations Section: Although the Authors state that limitations have been elaborated further, a more structured and clearly labelled subsection should be introduced, as a well-defined "Limitations" section.
-
The Authors provide a more detailed explanation of how k-means Clustering was handled and mention the use of the silhouette score. However, the response to the method’s appropriateness remains quite vague. The inherent limitations of k-means (e.g., equal variance assumption) are acknowledged in the reply but not truly and fully addressed in the manuscript. Consideration of alternative clustering methods or further discussion of this methodological limitation in the text would be advisable.
-
Wavelet Transform Discussion: The authors argue that further development of the WT section is unnecessary, as it is supplementary. While this Reviewer may accept the focus on Fourier analysis, the current treatment of WT remains quite superficial. Since it is included and cited, even a modest expansion would improve the manuscript’s balance and utility for readers.
-
The authors declined to incorporate the suggested references in their Literature Review, arguing a mismatch in sensing methods. However, the broader goal of the comment was to encourage a deeper contextualization of FRP sensing and analysis methods, also to other related applications, as it is now a bit too limited and restricted to a specific niche.
-
The Authors did expand the Conclusion Section, but the revised section still lacks sufficient depth in articulating implications and future directions. This section should more critically reflect on their meaning and how the study fits into the broader research landscape.
In light of these points, I believe the manuscript needs further major revisions.
Author Response
Dear Reviewer 3,
Thank you for your constructive comments which have enabled us to improve our manuscript. We have addressed your comments in the revised manuscript which we submit for further consideration for publication.
Below please find our response to your comments.
Several revisions have addressed earlier concerns to some extent, particularly in areas like figure clarity, limitations, and the addition of comparative tables. However, this Reviewer believes a further round of major revisions is necessary, as many points remain to be addressed:
- Limitations Section: Although the Authors state that limitations have been elaborated further, a more structured and clearly labelled subsection should be introduced, as a well-defined "Limitations" section.
Thank you for the comment. A limitations section has been clearly added as advised. Also limitations are discussed throughout the paper as they arise naturally as part of the critical discussion of the presented study.
2. The Authors provide a more detailed explanation of how k-means Clustering was handled and mention the use of the silhouette score. However, the response to the method’s appropriateness remains quite vague. The inherent limitations of k-means (e.g., equal variance assumption) are acknowledged in the reply but not truly and fully addressed in the manuscript. Consideration of alternative clustering methods or further discussion of this methodological limitation in the text would be advisable.
Thank you for comment regarding this, multiple measures have been taken to improve the validity of the results gathered in this section. The elbow method has now been included, as well as both visual and numerical evaluation for 4, 5, 6 and 7 clusters using the silhouette score. Additionally, an alternative form of clustering has been included, DBSCAN, which has automated designation of clusters and is fully unsupervised in this manner, in order to address some issues with k-means as well as serve to validate the results of this. These measures were completed for all samples, although only the one tensile sample is shown here as it is sufficient to demonstrate the discussed techniques.
3. Wavelet Transform Discussion: The authors argue that further development of the WT section is unnecessary, as it is supplementary. While this Reviewer may accept the focus on Fourier analysis, the current treatment of WT remains quite superficial. Since it is included and cited, even a modest expansion would improve the manuscript’s balance and utility for readers.
Thank you for the useful comment. On review of the manuscript the authors now understand the shortcomings of this approach. An expansion of the comments regarding the implications and use of WT within the context presented has been included to provide a more balanced comparison and more rounded discussion.
4. The authors declined to incorporate the suggested references in their Literature Review, arguing a mismatch in sensing methods. However, the broader goal of the comment was to encourage a deeper contextualization of FRP sensing and analysis methods, also to other related applications, as it is now a bit too limited and restricted to a specific niche.
Many thanks for the comments and apologies as the initial comment was misunderstood. Additional sources have been included and the introduction expanded in order to include areas where similar techniques which are presented here will be of use, for example alternative materials as an additional subsection in the introduction.
5. The Authors did expand the Conclusion Section, but the revised section still lacks sufficient depth in articulating implications and future directions. This section should more critically reflect on their meaning and how the study fits into the broader research landscape.
Many thanks for your comment. We have updated the conclusions section to be more in line with the expectations.
On behalf of the authors,
Professor Mayorkinos Papaelias